# Vital D: A modifiable occupational risk factor of UK healthcare workers

**James Phelan** [1☯], **Angukumar Thangamuthu**[1‡], **Srinivasagam Muthumeenal**[2‡],
**Kirsteen Houston**[3‡], **Mark Everton**[1‡], **Sathyanarayana Gowda**[1‡], **Jufen Zhang**[4‡],
**Rengarajan Subramanian** [1☯¤]*

1 Basildon Hospital, Mid and South Essex Foundation Trust, Basildon, United Kingdom, 2 Broomfield
Hospital, Mid and South Essex Foundation Trust, Broomfield, United Kingdom, 3 Southend Hospital, Mid and
South Essex Foundation Trust, Westcliff-on-Sea, United Kingdom, 4 School of Medicine, Faculty of Health
Education, Medicine and Social Care, Anglia Ruskin University, Cambridge, United Kingdom

☯ These authors contributed equally to this work.
¤Current address: Emergency Department, Basildon Hospital, Mid and South Essex Foundation Trust,
Basildon, United Kingdom
‡ AT, SM, KH, ME, SG and JZ also contributed equally to this work.
* rengarajan.subramanian@nhs.net

## Abstract

### Background

The role of Vitamin D in immune function is well reported with a growing evidence base linking low levels to poorer outcomes from infectious disease. Vitamin D deficiency and insufficiency are prevalent worldwide with healthcare workers identified as a known at-risk group. Here we aim to investigate serum Vitamin D levels in a UK population of front line healthcare workers and to promote the occupational risk.

### Methods

A cross-sectional study of 639 volunteers was conducted to identify the prevalence of Vitamin D deficiency and insufficiency amongst a population of front-line health care workers in the UK. Participant demographics and co-morbid factors were collected at the time of serum sampling for multivariate analysis.

### Results

Only 18.8% of the population had a normal vitamin D level greater than or equal to 75nmol/L. This is compared to Public Health England's (PHE) stipulated normal levels of 60% during winter. 81.2% had a level less than 75nmol/L, with 51.2% less than 50nmol/L and 6.6% less than 25nmol/L. For serum levels less than 25nmol/L, Asian ethnicity was more likely to have a vitamin D deficiency than non-asian (OR (95%CI): 3.81 (1.73-8.39), p = 0.001), whereas white ethnicity was less likely to have a vitamin D deficiency compared to non-white (OR (95%CI: 0.43 (0.20-0.83), p = 0.03). Other factors that contributed to a higher likelihood of lower-than-normal levels within this population included male sex, decreased age and not taking supplementation.

**Data Availability Statement:** Data cannot be shared publicly because potential identifiable data exists within the dataset. Data are available from the Mid and South Essex Research Institute (contact via mse.clinicalresearch@nhs.net) for

researchers who meet the criteria for access to confidential data.

**Funding:** - RS. - No grant number - The Mid and South Essex Hospitals Charity -The funders had no role in study design, data collection and analysis, decision to publish, or preparation of the manuscript.

**Competing interests:** The authors have declared that no competing interests exist.

## Conclusion

It is concluded that our population of healthcare workers have higher rates of abnormal vitamin D levels in comparison with the general UK population reported prevalence. Furthermore, Asian ethnicity and age 30 years and below are more at risk of vitamin D insufficiency and deficiency. This highlights an occupational risk factor for the healthcare community to consider.

## Introduction

The role of Vitamin D in immune function is well reported with a growing evidence base linking low levels to poorer outcomes from infectious diseases [1–3]. Further, Vitamin D is vital for calcium homeostasis and skeletal health, with risk of rickets, osteomalacia and osteopaenia in insufficient or deficient states [4]. In adults the clinical picture primarily presents with proximal musculoskeletal pain or weakness but may present with more diffuse effects or misdiagnosis of fibromyalgia or a depressive disorder. Front-line health care staff involved in direct care are regularly exposed to pathogens with a known increased occupational risk [5–7]. This risk is compounded through close contact or aerosol generating procedures in front-line workers [8]. The peak season for UK acute respiratory tract infection generally coincides with lower vitamin D concentrations during the winter months of December to February [9]. Furthermore, healthcare worker shift patterns can contribute to lower-than-normal exposure to natural sunlight [10]. The Covid-19 pandemic seen a disproportionate rate of infections in healthcare workers and the role of Vitamin D has been reported in both risk and treatment [6, 11–13].

In the UK, the Scientific Advisory Committee on Nutrition (SACN) defines vitamin D insufficiency and deficiency as $< 75 nmol/L$ and $25 nmol/L$, respectively. Lower levels of vitamin D affect at-risk groups and show seasonal variation in population studies [1]. Further, a body of evidence within the literature has shown people suffering from vitamin D deficiency to have higher rates of respiratory infections and poorer outcomes from sepsis [2, 3, 13, 14]. Acute setting healthcare workers are exposed to several factors putting them at risk of vitamin D deficiency and have been shown to have an increased prevalence compared to the general population [15]. A range of factors such as skin colour, age and the presence of pre-existing illness have also been associated with poorer outcomes from infectious disease [2, 11, 16, 17]. Of these, only vitamin D deficiency is modifiable and therefore target-able via occupational health policy.

The protective factor of Vitamin D has been suggested to come from its suppression of inflammatory cytokines released from macrophages. Lower levels of C-reactive protein, a molecule closely related to Interleukin-6, were recorded in patients with normal or high vitamin D levels [18]. For Covid-19, Vitamin D supplementation has been associated with reduced intensive care unit (ICU) admission, mechanical ventilation, and mortality [2, 12, 19]. Managing the release of IL-6 and other pro-inflammatory cytokines by exploiting metabolic alterations may effectively treat severe COVID-19 in the intensive care unit (ICU) [20]. In severe cases of sepsis, a Vitamin D level of less than 30nmol/L has been associated with increased mortality and longer hospital admission [14].

The primary objective of this study was to assess the prevalence of vitamin D deficiency in healthcare staff working in across an acute care hospital setting. Secondary objectives were to

assess the impacts of race, age, and vitamin D supplementation on the prevalence of deficiency within this cohort. Tertiary objectives were to inform occupational health policy.

A cross-sectional study was conducted to identify the prevalence of vitamin D deficiency among a population of front line health care workers in the UK. The consequential benefit of this analysis may prove a low-risk cost-effective measure of disease prevention and occupational risk factor reduction. Further, absence of staff due to work-related illness and well being may be addressed if strategies to risk assess and manage abnormal vitamin D levels are adopted.

## Materials and methods

### Ethical approval

The study was approved by NHS Health Research Authority and Health and Research Authority, Wales, U.K (reference id: 287417).

### Study design

A cross-sectional study with a sample size of 387 participants was needed to provide a two-sided 95% confidence interval with a precision of 0.05. An assumed prevalence of vitamin D deficiency of 40% among NHS staff with 35% missing values was required to give a minimum sample size of 596 participants in this cross-sectional study.

### Setting

Participants from three large NHS hospitals (Basildon and Thurrock University Hospital, Southend University Hospital, and Broomfield Hospital) of a single Trust (Mid-South Essex) in the East of England were included in this study. Participant recruitment, blood sampling and questionnaire completion was conducted in late January and early February, UK winter season, in the year 2021 during the COVID 19 pandemic.

### Participants

Health workers who were eighteen years and above of age and working for the National Health Service U.K. were included. Any workers below 18 years of age were excluded. A range of professions including doctors, nurses, HCA radiologists, porters and lab technicians were included. The study was advertised via email, educational seminars, and other local staff forums. Participation was voluntary.

### Measurements

Blood sampling to measure serum vitamin D levels and participant questionnaires were completed after obtaining consent across a total of 4 study days. Access 25(OH) Vitamin D Total assay, a two-step competitive binding assay using the UniCel DXI Immunoassay system, was used to measure total vitamin D levels in the samples.

### Quantitative variables

A Vitamin D level of greater than or equal to 75nmol/L was defined as normal. Serum levels of 75–50nmol/L, 50–25nmol/L and less than 25nmol/L were defined as low, insufficient, and deficient respectively.

### Descriptive variables

A paper questionnaire was used to collect participant variables as age, gender, ethnicity, co-morbidity's, vitamin supplementation and night-shift working pattern. This was collected at the time of blood sampling and inputted electronically to the anonymised data set.

### Statistical methods

Study data, including measured vitamin D levels, was collated electronically. Participants were given a unique identification number, and personal information was anonymised before analysis. A Separate team not involved in the analysis was tasked to communicate the measured vitamin D levels to participants. This was to make sure that any measured abnormal Vitamin D levels were reviewed by their corresponding general practitioner and acted upon.

Missing values of serum vitamin D levels were excluded from analysis. Missing questionnaire values were excluded as samples in the relevant section.

The study was carried out according to STROBE guidelines [21]. Continuous data's were summarized by mean (SD) or median (25th/75th percentiles) depending on the distribution of the data, while categorical data by counts (percent). The logistic regression models were used to identify the potential risk factors (such as age, sex, body mass index, race, and dietary supplementation) of the outcome (Vitamin D deficiency). The variables with a p-value < 0.2 found in the uni-variate analysis were included in the multi-variable logistic regression model. The results were presented as odds ratio (OR) with 95% confident intervals to show the strength of the association between risk factors and outcome. Models were checked for adequacy, and multi-colinearity was examined in them. A nominal level of 5% significance was used (two-tailed). The STATA statistical computer package was used to analyze the data.

## Results

### Participants

A total of 639 participants were recruited and sampled for serum Vitamin D. Seven vitamin D levels were not reported due to haemolysis in the sample. A total of 632 serum vitamin D levels were analyzed. Of those not all participants completed all sections of the accompanying questionnaire. This discrepancy resulted in variation in the total number analyzed for each category.

### Descriptive data

A total of 612 (95.7%) completed information of their gender, 626 (98%) of their ethnicity, 602 (94.2%) of their age and 605 (94.7%) of their night shift pattern. Of those that disclosed their age, the mean age was 54 (SD 28) years, the median was 48 (IQR 35–66). Of those that disclosed their gender identity, 80% were female.

### Outcome data: Serum vitamin D

The mean Vitamin D level was 54.32 (SD 27.96). The median vitamin D level was 48 (IQR 35–66) nmol/L with a minimum value of 15 nmol/L and a max value of 283 nmol/L. A total of 42 participants had a serum vitamin D level of less than 25nmol/L, representing 6.6%. A total of 282 participants had a level between 25–50nmol/L, representing 44.6%. A total of 189 had a level between 50–75nmol/L, representing 29.9%. The remaining 119 had a level greater than 75nmol/L, representing 18.8%. These results are displayed in Table 1. At classification of levels, 51.2% of the population had a level less than 50nmol/L, and 81.2% had a level less than

**Table 1. Serum vitamin D levels per categorisation).**

| Serum Vitamin D | < 25 | 25–50 | 50–75 | > 75 | Total |
|---|---|---|---|---|---|
| *Count* | 42 (6.7%) | 282 (44.6%) | 189 (29.9%) | 119 (18.8%) | 632 |

75nmol/L. The vitamin D sample distribution is outlined in Fig 1, showing a positively skewed distribution. Therefore, we infer that most participants were below the mean vitamin D level.

## Age

For vitamin D levels above 50 nmol/L, the mean (SD) of age was 44 (12) years, and the mean (SD) of age was 39 (11) years for vitamin D levels below 50 nmol/L. It appears that middle-aged and older participants had a higher percentage of vitamin D levels greater than 50 nmol/L. Conversely, younger participants (less than 50 years) had a lower percentage of taking vitamin D (Fig 2).

## Gender

For vitamin D levels above 75nmol/L, 20% of females and 17% of males recorded normal levels. However, 10% of males were deficient below the 25nmol/L level compared to 6% of females (Table 2).

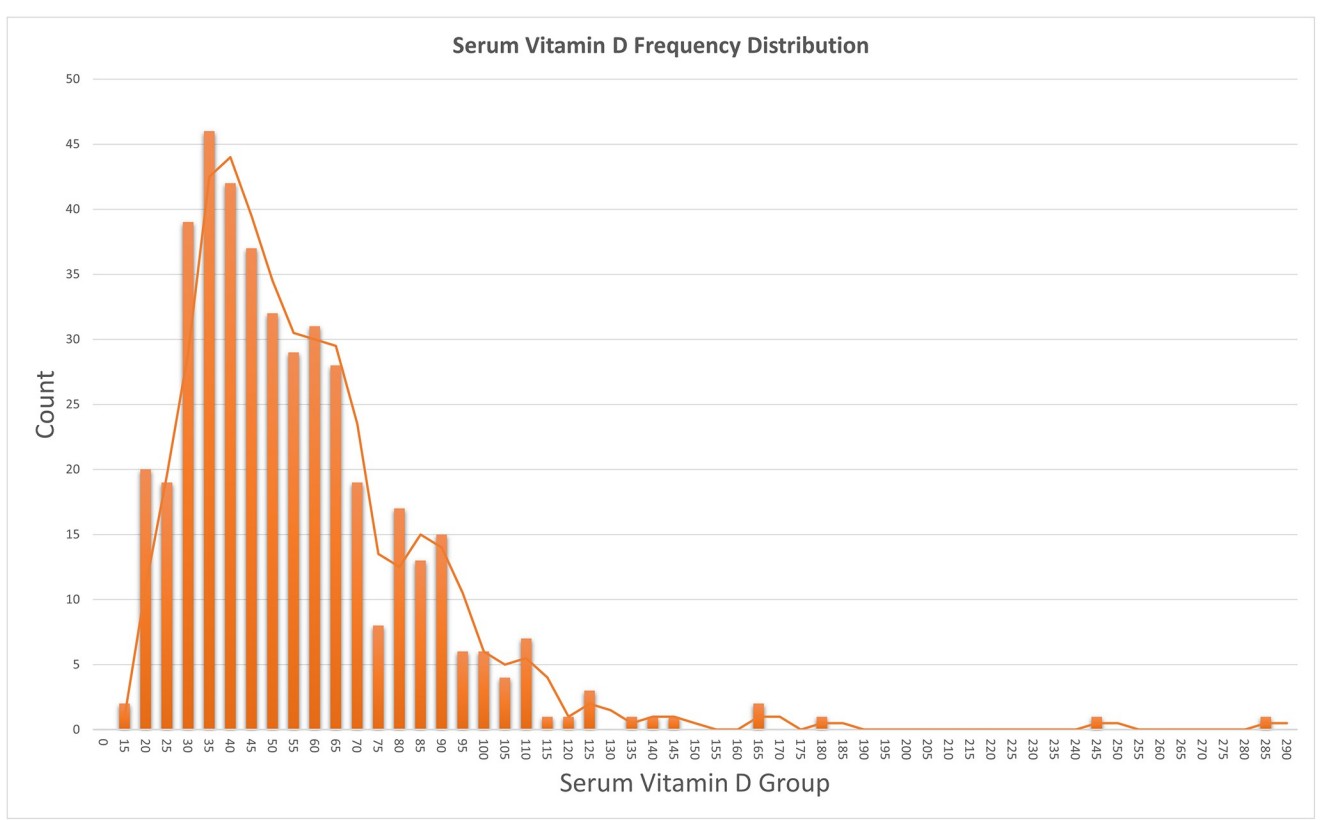

**Fig 1. Distribution.** Serum Vitamin D levels.

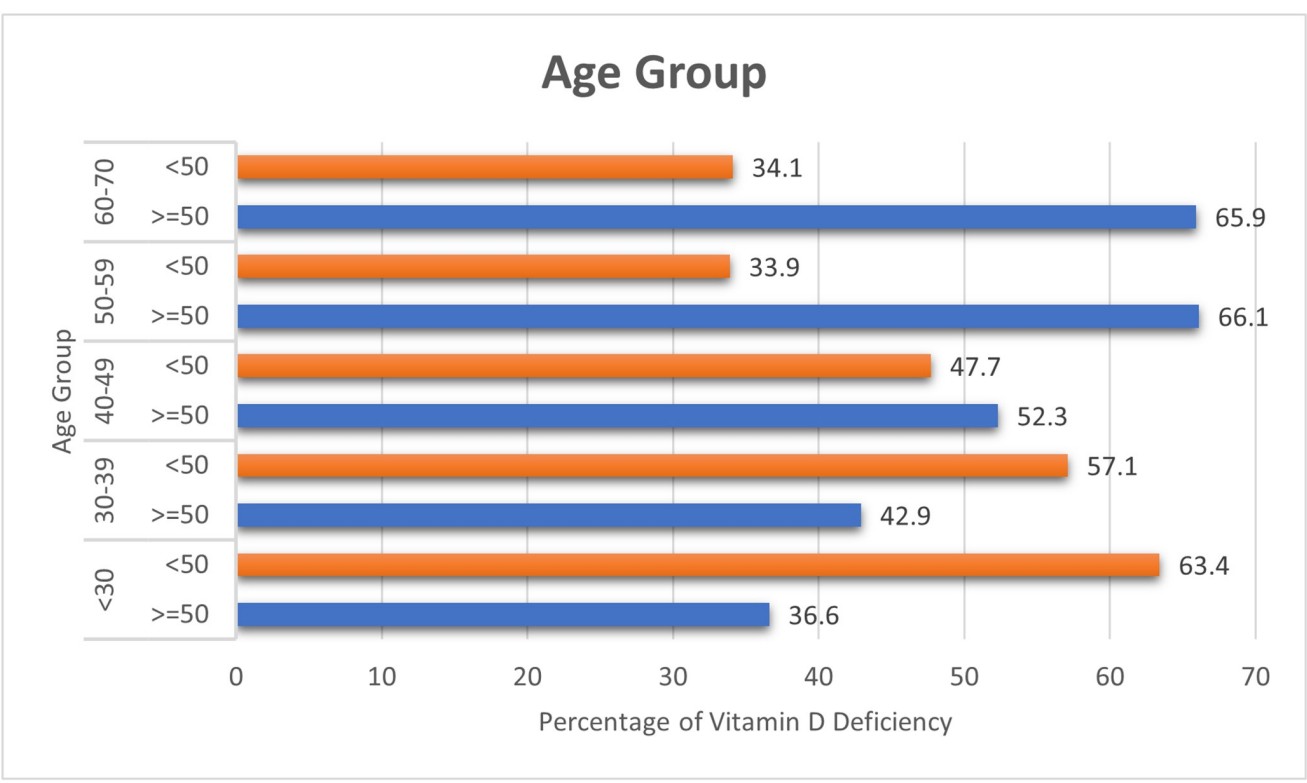

**Fig 2. Age.** Serum Vitamin D levels per age category.

## Ethnicity

Ethnicity data were collected to analyze the relationship between vitamin D levels and ethnicity, and to assess inclusion. A total of 626 participants recorded their ethnicity. The ratio of different ethnicity recorded is comparable to UK population census estimates. The serum Vitamin D level categorisation for different recorded ethnic groups is shown in Table 3. Fig 3 displays the proportion of each ethnicity group above and below a serum Vitamin D level of 50nmol/L. A higher proportion of results less than 50nmol/L was recorded in all ethnicity groups apart from White ethnicity.

## Supplementation

A total of 213 (33.7%) participants reported taking vitamin D supplementation, with 129 (60.6%) of this cohort still having levels recorded below 75nmol/L and 51 (23.9%) of levels below 50nmol/L. Three subjects taking supplementation still had a recorded level below

**Table 2. Serum vitamin D levels per gender category.**

| Serum Vitamin D | < 25 | 25–50 | 50–75 | > 75 | Total |
|---|---|---|---|---|---|
| *Female* | 28 (5.6%) | 214 (42.5%) | 161 (32%) | 100 (19.9%) | 503 |
| *Male* | 11 (10.4%) | 54 (50.9%) | 23 (21.7%) | 18 (17%) | 106 |
| *Unspecified* | 0 | 2 (66.7%) | 1 (33.3%) | 0 | 3 |

**Table 3. Serum vitamin D levels per ethnicity category.**

| Serum Vitamin D | < 25 | 25–50 | 50–75 | > 75 | Total |
|---|---|---|---|---|---|
| *White* | 18 (4.4%) | 183 (44.6%) | 137 (33.4%) | 72 (17.6%) | 410 |
| *Mixed, Multiple* | 1 (12.5%) | 6 (75%) | 0 | 1 (12.5%) | 8 |
| *Asian* | 17 (12%) | 58 (40.9%) | 33 (23.2%) | 34 (23.9%) | 142 |
| *Black, African, Caribbean* | 4 (7.7%) | 29 (55.8%) | 12 (23.1%) | 7 (13.5%) | 52 |
| *Other* | 2 (14.3%) | 6 (42.9%) | 3 (21.4%) | 3 (21.4%) | 14 |

25nmol/L. It was attempted to record dosage and regimen, including the onset of treatment date, but participants did not consistently know this information. Table 4 shows the number of people reporting supplementation in each of the categorisation of serum Vitamin D levels. As expected more people in the higher serum groupings were taking supplementation. Fig 4 shows the proportion of those taking supplementation separated by age group. The older age groups had a higher proportion taking supplementation than the younger groups.

## Other factors

BMI, Co-Morbidities, and Covid 19 antibody status were collected in the study questionnaire. However, the completion rates were inadequate, and their correlation to vitamin D levels was not analyzed further. Night shift working pattern was higher in the those with a Vitamin D level $< 50nmol/L$ than those with a Vitamin D level $> 50nmol/L$ (Table 5).

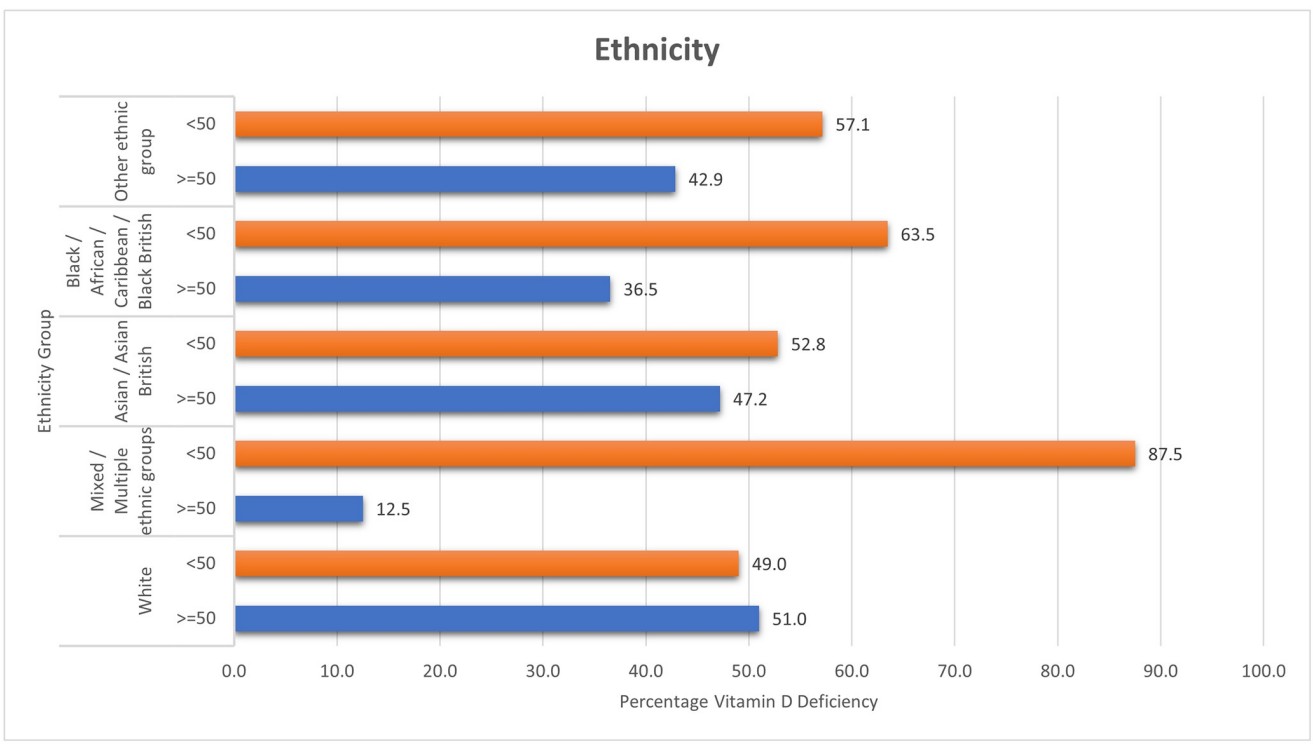

**Fig 3. Ethnicity.** Serum Vitamin D levels per ethnicity category.

**Table 4. Count taking vitamin D supplementation per serum vitamin D category.**

| Serum Vitamin D | < 25 | 25–50 | 50–75 | > 75 | Total |
|---|---|---|---|---|---|
| *Supplementation* | 3 (1.4%) | 48 (22.5%) | 78 (36.6%) | 84 (39.4%) | 213 |

### Association between vitamin D deficiency and independent variables

The results from the multi-variable logistic regression models showed that not taking vitamin D supplement (OR (95%CI): 0.17 (0.05–0.60), p = 0.005), age (OR (95%CI): 0.92 (0.88–0.96), p < 0.001) and non-white (0.43 (0.20–0.83), p = 0.03) were independently associated with vitamin D deficiency (below 25 nmol/L). We also noted that, compared to other ethnicity's, Asian ethnicity participants (3.81 (1.73–8.39), p = 0.001) were associated with an increased risk of vitamin D deficiency if excluding the White variable in the model. In addition, when the vitamin D deficiency definition by a cut-off point of 50 nmol/L was applied, compared to females, males were more likely to have a vitamin D deficiency. These results are displayed in Table 6

### Discussion

A cutoff of 50nmol/L has been used in the UK to diagnose insufficient vitamin D levels and guide management. Within this cohort of healthcare workers, 51.2% have been classified as having insufficient levels based upon NICE guidance. Of them, 42 (6.6%) participants had

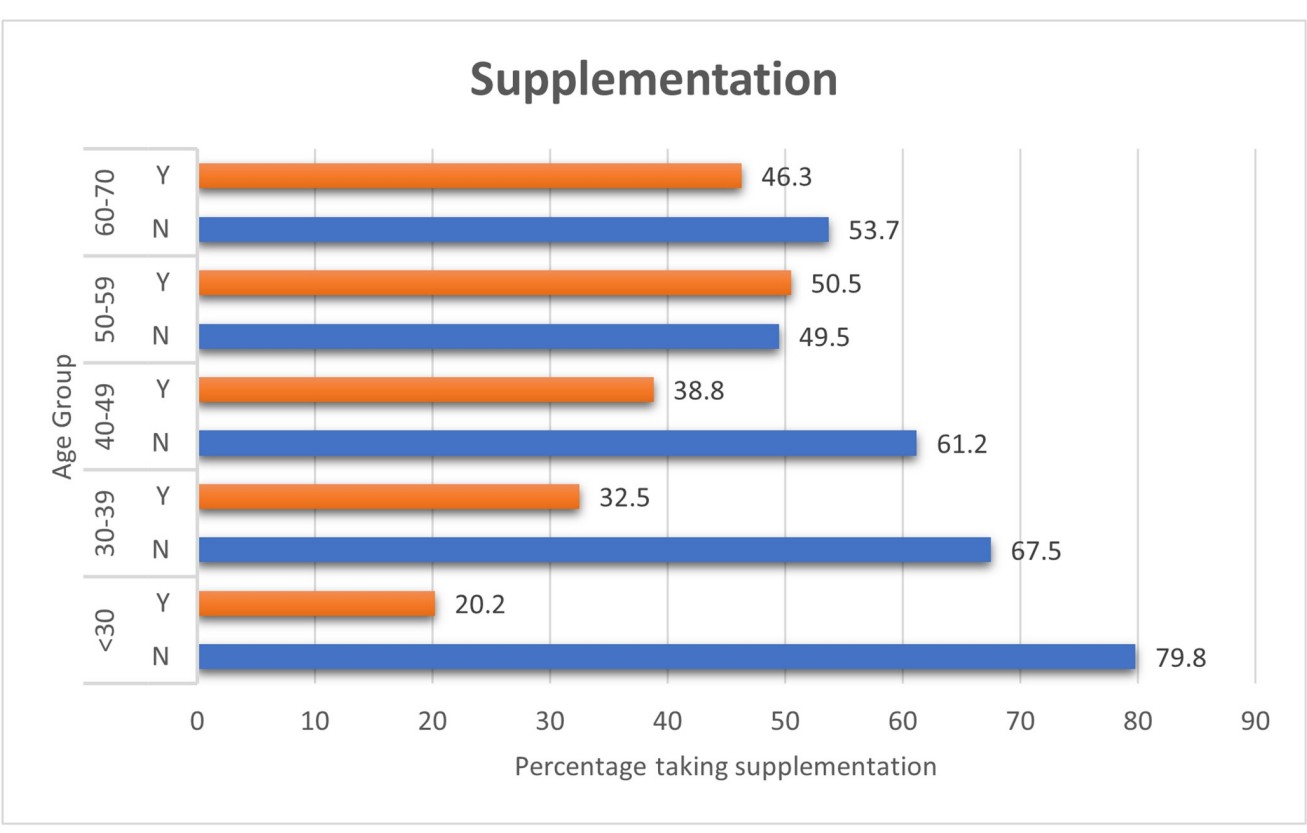

**Fig 4. Supplementation.** Vitamin D supplementation per age category.

**Table 5. Night shift working pattern per serum vitamin D category.**

| Serum Vitamin D | < 25 | 25–50 | 50–75 | > 75 | Total |
|---|---|---|---|---|---|
| *Night–shifts* | 31 (10.1%) | 146 (47.4%) | 86 (27.9%) | 45 (14.6%) | 308 |
| *No–Night–shifts* | 8 (2.7%) | 124 (41.8%) | 98 (33%) | 67 (22.6%) | 297 |

vitamin D levels that are considered to be severely low ($< 25 nmol/L$). A total of 119 (18.8%) participants alone were considered to have a normal vitamin D level. In contrast, the UK general population has an estimated prevalence of 40% during winter months [10]. Given the context of previous works highlighting an increased prevalence of vitamin D deficiency in healthcare workers, these results further contribute to the literature base.

We have further explored multiple risk factors that contributed to a higher likelihood of lower-than-normal levels, which include male sex, decreased age, and not taking supplementation. It was considered that the effects of older age might be due to an increased proportion of those receiving supplementation. However, regression modeling showed that both the supplemented and non-supplemented cohorts had lower predicted values of vitamin D deficiency with increasing age; though, a lesser gradient was observed in the supplemented group. These findings contradict other studies and guidance correlating older age with lower vitamin D levels. The contrary nature of this outcome may highlight the impact of indoor shift working and night shift patterns experienced in this younger cohort as supported in other works.

Ethnicity is another well-established factor for vitamin deficiency. Our results showed that Asian ethnicity had a positive correlation, but the multi-variate analysis did not elicit significant relationships between ethnicity and Vitamin D deficiency in this cohort. These findings suggest that being a healthcare worker alone is highly contributory.

The study was limited by a small reduction of participants due to haemolysis of blood sampling. Associated factor analysis was impacted by variable completion of participant questionnaires with missing values being excluded from these parts of the analysis. Participant confidence in Vitamin D supplementation dosage was not consistent and identified at time of completion, therefore, Vitamin D supplementation was considered as a binary (yes/no) variable.

**Table 6. Univariate and multivariate logistic regression models (using a vitamin D cut-off level of 25).**

| | Univariate | | | Multivariate | | |
|---|---|---|---|---|---|---|
| | OR (95%CI) | z-value | p-value | OR (95%CI) | z-value | p-value |
| *Age(yrs)* | 0.92(0.88–0.95) | -4.36 | < 0.001 | 0.92(0.88–0.96) | -3.67 | < 0.001 |
| *Sex(Female)* | 0.51(0.24–1.06) | -1.81 | 0.07 | - | - | – |
| *BMI > 30* | 1.33(0.56–3.19) | 0.65 | 0.52 | - | - | - |
| *White–ethnicity* | 0.37(0.19–0.70) | -3.08 | 0.002 | 0.43 (0.20–0.93) | -2.14 | 0.03 |
| *Asian–ethnicity* | 2.47(1.29–4.72) | 2.74 | 0.006 | 3.81(1.73–8.39) | 3.33 | 0.001 |
| *Black–ethnicity* | 1.17(0.40–3.43) | 0.30 | 0.77 | - | - | - |
| *Mixed–ethnicity* | 2.01(0.24–16.76) | 0.65 | 0.52 | - | - | - |
| *Other–ethnicity* | 2.38(0.51–10.99) | 0.27 | 0.27 | - | - | - |
| *Supplementation* | 0.14(0.04–0.45) | -3.28 | 0.001 | 0.17(0.05–0.60) | -2.78 | 0.005 |

The model was adjusted for age, sex, White ethnicity and vitamin D supplementation. Asian ethnicity was not included as there was a highly correlation between White and Asian ethnicity's. Asian was significantly associated with vitamin D deficiency excluding White ethnicity.

Vitamin D has been implicated in the treatment and prognosis of a range of infectious diseases. Therefore, maintaining an optimal level among front-line staff is a directly modifiable element that can be easily cost-effective. Other work has identified a study protocol for a randomised control trial to investigate the use of Vitamin D supplementation to reduce the number of symptomatic positive COVID-19 cases in healthcare workers. This study has demonstrated the prevalence of vitamin D deficiency/insufficiency in a subset of front-line UK, NHS healthcare workers, which may put them at risk of infection with an unfavorable outcome. Further work would be required to consider how these findings would generalise to other populations. We consider these findings could be adopted to inform occupational health policy in this workforce alongside other measures such as hand hygiene, effective personal protective equipment (PPE) usage and vaccination strategies.

## Conclusion

It is concluded that our population of healthcare workers have higher rates of abnormal vitamin D levels in comparison with the general UK population reported prevalence. Furthermore, Asian ethnicity and age 30 years and below are more at risk of vitamin D insufficiency and deficiency. This highlights an occupational risk factor for the healthcare community to consider.

## Supporting information

**S1 Checklist. STROBE statement—Checklist of items that should be included in reports of observational studies.**
(DOCX)

## Author Contributions

**Data curation:** James Phelan, Angukumar Thangamuthu, Kirsteen Houston, Mark Everton, Jufen Zhang.

**Formal analysis:** James Phelan, Jufen Zhang.

**Funding acquisition:** Rengarajan Subramanian.

**Investigation:** James Phelan, Angukumar Thangamuthu, Srinivasagam Muthumeenal, Kirsteen Houston, Mark Everton, Sathyanarayana Gowda, Jufen Zhang, Rengarajan Subramanian.

**Methodology:** James Phelan, Mark Everton, Jufen Zhang, Rengarajan Subramanian.

**Project administration:** James Phelan, Rengarajan Subramanian.

**Resources:** Srinivasagam Muthumeenal, Kirsteen Houston, Rengarajan Subramanian.

**Supervision:** Rengarajan Subramanian.

**Validation:** James Phelan, Jufen Zhang.

**Visualization:** James Phelan, Jufen Zhang.

**Writing – original draft:** James Phelan, Mark Everton, Jufen Zhang.

**Writing – review & editing:** James Phelan, Angukumar Thangamuthu, Srinivasagam Muthumeenal, Kirsteen Houston, Mark Everton, Jufen Zhang, Rengarajan Subramanian.

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
