## [Decision Letter · Decision Letter 0]

12 Sep 2023

PONE-D-23-08403Vital D: A Modifiable Occupational Risk Factor for Healthcare WorkersPLOS ONE

Dear Dr. Subramanian,

Thank you for submitting your manuscript to PLOS ONE. After careful consideration, we feel that it has merit but does not fully meet PLOS ONE’s publication criteria as it currently stands. Therefore, we invite you to submit a revised version of the manuscript that addresses the points raised during the review process.

We look forward to receiving your revised manuscript.

Kind regards,

Nasser Hadal Alotaibi

Academic Editor

PLOS ONE

Journal Requirements:

"The work was supported by funds from Mid and South Essex Hospitals charity. We also would like to thank research and development department Mid and South Essex hospitals U.K. for their support and junior doctors of Mid and South Essex hospitals who volunteered to assist in sample collection."

"- RS.

- No grant number

- The Mid and South Essex Hospitals Charity

-The funders had no role in study design, data collection and analysis, decision to publish, or preparation of the manuscript"

5. Please amend your authorship list in your manuscript file to include author "A. Thangamuthu1‡, S. Muthumeenal2‡, K. Houston3‡, M. Everton1‡, S. Gowda 1‡, J.Zhang"

Reviewers' comments:

Reviewer's Responses to Questions

**Comments to the Author**

1. Is the manuscript technically sound, and do the data support the conclusions?

Reviewer #1: Yes

Reviewer #2: Partly

Reviewer #3: Yes

2. Has the statistical analysis been performed appropriately and rigorously? 

Reviewer #1: Yes

Reviewer #2: Yes

Reviewer #3: Yes

3. Have the authors made all data underlying the findings in their manuscript fully available?

Reviewer #1: Yes

Reviewer #2: Yes

Reviewer #3: Yes

4. Is the manuscript presented in an intelligible fashion and written in standard English?

Reviewer #1: Yes

Reviewer #2: Yes

Reviewer #3: Yes

5. Review Comments to the Author

Reviewer #1: Title: add UK Healthcare workers

Introduction, paragraph 1: Clarify winter months in the UK (i.e. Canada's winters last at least oct-march/april); expand on signs and symptoms of vit D def

Line 35 has an upside down exclamation mark.

Study population: Clarify the healthcare workers' positions i.e. nurses, docs, porters, respiratory therapists, etc

Line 108: missing period

All tables: I would place percentages behind the value i.e. 42 (6.6%)

Future considerations: Sample same group in winter months and then again at peak summer months

The article would be stronger if the authors expanded on the current literature, such as Sowah et al's 2017 article that was only briefly mentioned. Another suggstion: "Prevention of COVID-19 with oral vitamin D supplemental therapy in essential healthcare teams (PROTECT): protocol for a multicentre, triple-blind, randomised, placebo-controlled trial" by Ducharme et al (2023).

Reviewer #2: The authors aimed to investigate deficiency and insufficiency of serum Vitamin D levels in 639 volunteered front-line healthcare workers in the UK.

Introduction:

Other factors, such as hand hygiene, personal protective equipment (PPE) use, and vaccination status, also play crucial roles in infection prevention for front-line healthcare workers, which must be mentioned in the introduction or discussion section.

Materials and methods:

This study was carried out at Mid and South Essex NHS Foundation Trust, and the population does not represent UK front-line healthcare workers. Please modify it appropriately in different parts of the manuscript.

The first subsection of the Materials and methods is called study design. However, it is more about sample size calculation and justification.

Relevant dates, including periods of recruitment, exposure, and data collection, should be mentioned.

It is not well clear that the inclusion criteria were for workers at the “Mid and South Essex NHS Foundation Trust” or the National Health Service UK.

The size of participants from the sampling frame and the proportion who agreed to take part in this study are not mentioned.

Different subsections of materials and methods are mixed up together. Based on the STROBE statement, it could be good to have the following subsections: Study design, Setting, Participants, Variables, Measurements, Data sources, Bias, Study design, Quantitative variables, and Statistical methods.

Results:

Following the STROBE statement for writing and presenting the results could be more appropriate.

Adding the percentages to the counts in Table 1 makes it more informative, and it is possible to delete the two sentences in Lines 119-122.

It is similar to other tables, and combining Tables 1, 2, 3, 4 and 5 could be reasonable.

Marking the mean and median in Figure 1 could be informative.

Discussion:

The authors concluded that vitamin D deficiency/insufficiency puts the front-line UK NHS healthcare Workers at risk of infection with an unfavourable outcome beyond this study's design, measurements and analysis.

Reviewer #3: Well described manuscript with focus on the Health Care Workers (HCW). As vitamins are the vital one for the healthy living, much more studies are essential on this area of research in the upcoming years for the future benefits.

6. PLOS authors have the option to publish the peer review history of their article (what does this mean?). If published, this will include your full peer review and any attached files.

Reviewer #1: No

Reviewer #2: No

Reviewer #3: No

---

## [Author Response · Author response to Decision Letter 0]

19 Nov 2023

Specific responses that respond to each point raised by the academic editor and reviewer(s).

1. The manuscript has been confirmed to meet PLOS ONE's style requirements to the best of our knowledge.

2. Funding-related text has been removed from the manuscript. The funding statement does not need to be changed.

3. The grant information has been inserted within the funding information section to offer further clarification. No specific grant number exists and managed internally within the hospital trust.

4. The data remains available upon request due to potentially sensitive personal information.

a. Dr Rengarajan Subramanian remains the data custodian for this work and contactable via email (rengarajan.subramanian@nhs.net).

b. Data not uploaded for the above reasons.

5. Authorship list in manuscript in-keeping with request.

6. References: reference 23 has been included after review due to added content.

Reviewer #1 comments:

• Title: add UK Healthcare workers

- Change made

• Introduction, paragraph 1: Clarify winter months in the UK (i.e. Canada's winters last at least oct-march/april); expand on signs and symptoms of vit D def

- Clarified winter months and expanded on signs & symptoms

• Line 35 has an upside down exclamation mark.

- Removed

• Study population: Clarify the healthcare workers' positions i.e. nurses, docs, porters, respiratory therapists, etc

- We have expanded upon this in the manuscript

• Line 108: missing period

- Added

• All tables: I would place percentages behind the value i.e. 42 (6.6%)

- We have added these to tables were relevant

• Future considerations: Sample same group in winter months and then again at peak summer months

- No change made to manuscript

• The article would be stronger if the authors expanded on the current literature, such as Sowah et al's 2017 article that was only briefly mentioned. Another suggstion: "Prevention of COVID-19 with oral vitamin D supplemental therapy in essential healthcare teams (PROTECT): protocol for a multicentre, triple-blind, randomised, placebo-controlled trial" by Ducharme et al (2023).

- We have added to the manuscript based upon this and referenced under [23].

Reviewer #2 comments:

• Introduction: Other factors, such as hand hygiene, personal protective equipment (PPE) use, and vaccination status, also play crucial roles in infection prevention for front-line healthcare workers, which must be mentioned in the introduction or discussion section.

- Content added to manuscript

• Materials and methods: This study was carried out at Mid and South Essex NHS Foundation Trust, and the population does not represent UK front-line healthcare workers. Please modify it appropriately in different parts of the manuscript.

- Manuscript amended to account for this by referencing a sample of UK front-like healthcare workers.

• The first subsection of the Materials and methods is called study design. However, it is more about sample size calculation and justification.

- The materials and methods have been restructured based upon this feedback.

• Relevant dates, including periods of recruitment, exposure, and data collection, should be mentioned.

- Within manuscript

• It is not well clear that the inclusion criteria were for workers at the “Mid and South Essex NHS Foundation Trust” or the National Health Service UK.

- This has been clarified within manuscript.

• The size of participants from the sampling frame and the proportion who agreed to take part in this study are not mentioned.

- Participants section amended to clarify.

• Different subsections of materials and methods are mixed up together. Based on the STROBE statement, it could be good to have the following subsections: Study design, Setting, Participants, Variables, Measurements, Data sources, Bias, Study design, Quantitative variables, and Statistical methods.

- The materials and methods have been restructured based upon this feedback

• Results: Following the STROBE statement for writing and presenting the results could be more appropriate.

- Results section has been reorganised and amended to align with STROBE statement

• Adding the percentages to the counts in Table 1 makes it more informative, and it is possible to delete the two sentences in Lines 119-122.

- We have added these to tables were relevant

• It is similar to other tables, and combining Tables 1, 2, 3, 4 and 5 could be reasonable.

- This has not been changed as it was felt the combined tables was overwhelming when trialled.

• Marking the mean and median in Figure 1 could be informative.

- This has been added

• Discussion: The authors concluded that vitamin D deficiency/insufficiency puts the front-line UK NHS healthcare Workers at risk of infection with an unfavourable outcome beyond this study's design, measurements and analysis.

- Corrected within manuscript

Reviewer #3 comments:

• Can you specify the reason for Vitamin D deficiency particularly among the front line healthcare workers?

- This would be beyond the remit of this study to conclude this.

• Does any of the study population had COVID infection during the study period?

- Active COVID-19 was not tested for during the study. However, it can be assumed that anybody with active COVID-19 at the time of the study would have been self-isolating and therefore not in attendance or included upon study days.

• Does any of the study population was already on calcium supplementation?

- We established vitamin D supplementation during this study but did not collect data upon calcium supplementation.

• Does any of the study population had CKD or other chronic illness?

- A range of chronic illnesses were questioned upon within the study questionnaire including CKD, hypertension, diabetes, heart disease, chronic lung disease. A small number (n=3) reported a background of CKD.

---

## [Decision Letter · Decision Letter 1]

10 Dec 2023

Vital D: A Modifiable Occupational Risk Factor of UK Healthcare Workers

PONE-D-23-08403R1

Dear Dr. Subramanian,

We’re pleased to inform you that your manuscript has been judged scientifically suitable for publication and will be formally accepted for publication once it meets all outstanding technical requirements.

Kind regards,

Nasser Hadal Alotaibi

Academic Editor

PLOS ONE

Additional Editor Comments (optional):

Reviewers' comments:

Reviewer's Responses to Questions

**Comments to the Author**

1. If the authors have adequately addressed your comments raised in a previous round of review and you feel that this manuscript is now acceptable for publication, you may indicate that here to bypass the “Comments to the Author” section, enter your conflict of interest statement in the “Confidential to Editor” section, and submit your "Accept" recommendation.

Reviewer #2: All comments have been addressed

2. Is the manuscript technically sound, and do the data support the conclusions?

Reviewer #2: Yes

3. Has the statistical analysis been performed appropriately and rigorously? 

Reviewer #2: Yes

4. Have the authors made all data underlying the findings in their manuscript fully available?

Reviewer #2: Yes

5. Is the manuscript presented in an intelligible fashion and written in standard English?

Reviewer #2: Yes

6. Review Comments to the Author

Reviewer #2: Thanks for the ammendments.

The authors have adequately addressed your comments raised in a previous round of review and I feel that this manuscript is now acceptable for publication

7. PLOS authors have the option to publish the peer review history of their article (what does this mean?). If published, this will include your full peer review and any attached files.

Reviewer #2: No

---

## [Editor Report · Acceptance letter]

31 Jan 2024

PONE-D-23-08403R1 

PLOS ONE

Dear Dr. Subramanian, 

I'm pleased to inform you that your manuscript has been deemed suitable for publication in PLOS ONE. Congratulations! Your manuscript is now being handed over to our production team.

Kind regards, 

on behalf of

Dr. Nasser Hadal Alotaibi 

Academic Editor

PLOS ONE